# A NEAT quantum error decoder

**Hugo Théveniaut[1,2] and Evert van Nieuwenburg[2]**

**1** Laboratoire de Physique Théorique, IRSAMC, Université de Toulouse,
CNRS, UPS, 31062 Toulouse, France
**2** Niels Bohr International Academy, Niels Bohr Institute, University of Copenhagen,
Universitetsparken 5, 2100 Copenhagen, Denmark

## Abstract

We investigate the use of the evolutionary NEAT algorithm for the optimization of a policy network that performs quantum error decoding on the toric code, with bitflip and depolarizing noise, one qubit at a time. We find that these NEAT-optimized network decoders have similar performance to previously reported machine-learning based decoders, but use roughly three to four orders of magnitude fewer parameters to do so.

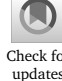
# 1 Introduction

Over the recent years, machine learning techniques for quantum physics have become more and more commonplace [1, 2]. These techniques provide a rather different paradigm to solving hard problems than traditional algorithms do. Instead of explicitly constructing algorithms –taking special care of all possible scenarios that may occur, manually– machine learning techniques are capable of learning what to do autonomously.

It is common to categorize these learning algorithms in three classes, namely those of i) supervised learning and ii) unsupervised learning, where information is extracted from example data, and that of iii) reinforcement learning (RL), where the learner has the ability to interact with the problem and receive feedback in the form of a reward. Each of these three classes have seen applications to various physics problems. In particular, supervised learning and reinforcement learning have proved potentially useful for quantum error correction on 2D stabilizer codes [3]. The supervised approach is represented by Refs. [4–9], where a dataset of errors and valid corrections is used to extract what the most likely correction to be performed is. Compared to hard-coded decoding algorithms for stabilizer codes, these machine learning based decoders are more like maximum-likelihood decoders [10, 11] than, for instance, the minimum weight perfect matching (MWPM) algorithm that looks for the lowest energy correction [12]. In the RL based approach, the decoding problem is formulated as a move-based single player game [13–16]: the player proposes a local correction, receives back the new state of the code and wins whenever the error-free state of the code is restored correctly. These machine learning decoders are mostly limited to small sizes (albeit comparable to a possible realistic experimental implementation), though scalability through hybrid approaches is a promising research direction [17]. The flexibility of the machine learning approach has the interesting prospect of being useful for the decoding of realistic codes in which qubits are not all identical, suffer from distinct error rates and in which measurements are faulty [18].

This work falls into the class of reinforcement learning approaches. The previous contributions in this direction mentioned above employed deep Q-learning [19], where a deep network is used to approximate the so-called *Q*-function from which the policy (i.e. which correction to do given the state of the system) can be found. The number of parameters required in deep *Q*-function networks can be extremely large however, making training a computationally intensive step that requires the back-propagation algorithm for the network gradient's evaluation and fast network evaluation with GPUs.

In this work we investigate a new type of setup where i) a neural network approximates directly the agent's policy and ii) optimization is performed with an evolutionary algorithm, namely that of neuro-evolution of augmenting topologies (NEAT) [20]. The novelty of NEAT lies in its ability to not only optimize the weights of neural networks, but also the architecture: it allows for nodes or connections between nodes to be added during optimization (see Fig. 1). This is made possible by a clever encoding of neural networks in terms of a genome, enabling a meaningful way to 'cross-over' two networks. We find that the NEAT algorithm is easily capable of finding a decoding strategy that performs similarly to MWPM (as do the previous RL decoders) on the toric code.

There are several advantages to our approach, compared to using *Q*-learning. First, since NEAT automatically optimizes the network architecture, the problem of manually designing and tweaking the model (i.e. how many layers and neurons) is no longer relevant. Second, it is a gradient-free optimization technique that is possibly faster than back-propagation [21], and highly parallelizable (discussed further in Section 2). Third, due to the genome encoding the networks, a straightforward 'genome transplant' allows us to use a trained network from smaller system sizes as a starting point for larger ones (see Appendix B). Last, we find that the networks found by NEAT are three to four orders of magnitude smaller than the equivalent

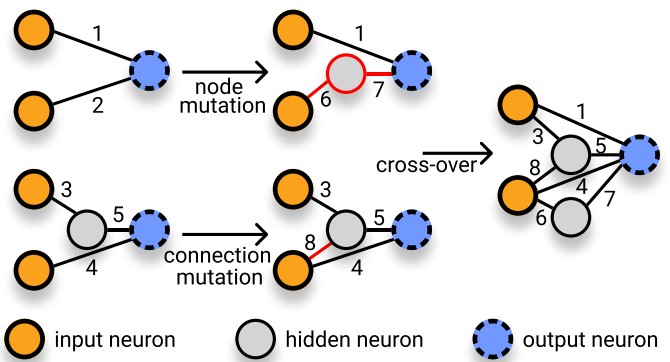

Figure 1: In the NEAT algorithm, a population (set) of neural networks undergo a series of mutation, selection and reproduction processes repeated over several generations (optimization steps). The mutations involve randomly changing the values of the network parameters (with some probability) as well as randomly modifying the architecture of the network by adding/removing hidden neurons or connections between neurons (with some probability). For structural mutations, hidden neurons are always added by splitting an existing connection, and the set of input and output neurons are left untouched. Each generation also selects part of the population for crossover, without which the optimization would be purely reliant on random mutations. Each connection is assigned a unique number (a "genome marker"), which enables a cross-over procedure where two networks of distinct architectures can be meaningfully combined into a new network. See Appendix A for more details.

$Q$-networks. This may be important in applications of these networks, since smaller networks can be evaluated faster.

The rest of this paper is structured as follows. We continue with the introduction of the NEAT algorithm for optimizing a policy network. We then introduce the toric code as a move-based single player game, so that an RL agent (the policy network) can be trained on the decoding task. Last up is a presentation of the results, and a discussion on some advantages and disadvantages of our approach compared to previous literature. Future enhancements of this approach are mentioned at the very end.

## 2 The NEAT Algorithm

In standard evolutionary strategies, the optimized solution is not found using a gradient based method. Instead, a population of candidate solutions is evolved over several evolutionary steps called generations according to heuristics inspired by biological evolution. Each individual in the population is assigned a fitness (a figure of merit for how well it is doing at solving the task), and optimization is then done each generation (i) via random mutations of individuals, (ii) selection and (iii) reproduction of the best performing individuals via crossover between them that direct the search towards the best fitness. Recent example applications of evolutionary strategies related to physics are that of combinatorial optimization problems [22] and the automated discovery of new semiconductor materials [23].

In addition to the mutations that affect the network parameters, structural mutations come in the form of adding/removing weights and hidden neurons, directly modifying the network topology, as shown in Fig. 1. Crossover events between networks of different topologies is not

---

**Algorithm 1:** The NEAT algorithm for decoding

Initialize a new population of trivial networks
**for** *num_generations* **do**
    **foreach** *network N in the population* **do**
        Play $N_g$ games (Algorithm 2)
        Fitness = number of won games / $N_g$
        Mutate randomly with probability $p$
    **end**
    Move top individuals to the new generation
    Cross-over top individuals per species
**end**
Return network with the highest fitness

---

straightforward, and the essential part of the NEAT algorithm is to enable this using an encoding of network topology in a genome [20]. The genome encodes neuron types (input, hidden, output), neuron biases, their connections (weights) and whether or not a given connection is enabled. Appendix A discusses this in more detail.

NEAT uses two further tweaks to the standard evolutionary algorithm. First, to counter the fact that random mutations will decrease the fitness at first, although they may be the beginning of a branch of better individuals in the long-term, similar individuals in the population are grouped together and are evolved separately. This mechanism enables the protection of innovation through *speciation*.

Second, evolution starts with trivial networks containing only the input and output neurons. New architectural components are introduced by mutation and crossover and tested in isolation thanks to the speciation mechanism; after some time only the most fit species survive. As a result, the complexity increases *only when necessary* and results in solutions of minimal complexity.

Networks optimized using NEAT show excellent performance on different control tasks benchmarks [20, 24] and are of very small complexity compared to their back-propagation trained equivalents. The parallelization of the algorithm is straightforward since the fitness of each network in the population can be evaluated independently, on independent games.

## 3 Decoding on the toric code as a game

In a nutshell, the toric code represents two *logical* qubits in the fourfold degenerate ground-states of a 2D periodic square lattice of *physical* qubits [25]. The size of the lattice is referred to as the code distance $d$ [1]. The system is governed by the Hamiltonian

$$H = -\sum_{\text{plaquette}} P - \sum_{\text{star}} S \,, \tag{1}$$

where the stabilizers $P$ ($S$) are products of 4 Pauli $Z$ ($X$) operators around a plaquette (star), see Figure 2. The groundstate space is spanned by the states for which all plaquette and star operators have eigenvalue $+1$.

If a single bitflip/phaseflip (Pauli $X/Z$) error occurs on a physical qubit, the two adjacent plaquette/star operators will measure $-1$ and will show a syndrome (indicated by the (orange) circles in Fig. 2). Further errors can move these syndrome endpoints around, forming an

---

[1]The code distance is an important quantity that indicates the smallest possible number of physical qubit errors that would cause a logical error.

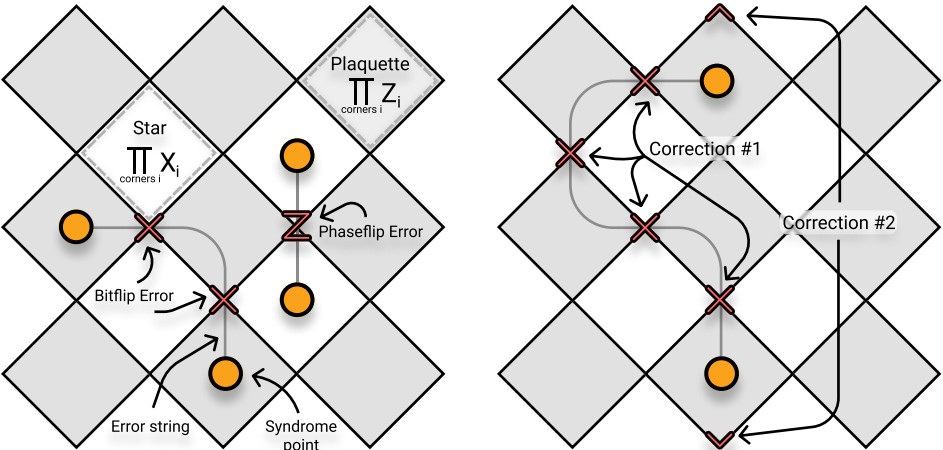

Figure 2: **Left Panel**: The toric code essentials. Qubits live on the vertices of the grid, with the plaquette/star operators then being formed by the (darker/lighter) squares. Bitflip errors and phaseflip errors, indicated by the $X$ and $Z$ operations on the vertices leave behind error strings with syndrome points at their endpoints. **Right Panel**: An example error string and two possible corrections that both would remove the syndrome. Correction #1 would exactly undo the errors, but correction #2 would introduce a non-trivial loop resulting in a logical error.

error string. When two syndrome points meet on a plaquette, $P$ again has value $+1$ and the syndrome disappears (similarly for $S$). This is important, because it means that the system can be brought back to the groundstate space by either perfectly undoing all the errors, or by closing error strings into trivial (contractable) loops. Only if an error string is closed by looping around the periodic boundaries (forming a non-contractable loop), does the logical state encoded in the groundspace incur an error. As long as we can correct the physical qubits before such a non-trivial loop forms, the logical qubit can be protected.

A single game of the toric code is then played as follows. The system starts out in the groundstate with no syndrome, on which random errors are introduced with given probabilities $p_{\text{error}}$. We will consider below two types of error models: (i) uncorrelated noise where Pauli $X$ operators are applied with probability $p_{\text{error}}$ on each site and (ii) depolarizing noise where either Pauli $X$, $Y$ or $Z$ operators are inserted with probability $p_{\text{error}}$. The game then progresses by making one move at a time, acting with Pauli operators on qubits to move the syndrome points around in an attempt to merge them. Finishing the game consists of acting on the physical qubits one-by-one until no more syndrome points are left. At that point, the total error strings –including the original errors introduced at the start– are evaluated and if no logical error is present, the game is won.

We implemented this game as a reinforcement learning problem using the OpenAI Gym [26] framework, and made it publicly available as part of SciGym [27]. Using this environment we use the NEAT algorithm to optimize a policy network $N(s) \rightarrow a$ that takes as input the state $s$ of the game and outputs the probability to take action $a$ (which qubit to act on with which Pauli operator) [2]. The state $s$ of the game is taken to be the current measurements of the stabilizer operators $P$ and $S$ (amounting to $2d^2$ values), meaning that the input has no memory of the past. As pointed out in Ref. [15], this implements exponential compression of information.

In principle, the entire action space of possible moves consists of each qubit and which

---

[2]The difference with Q-learning lies in that the agent's policy $\pi(a|s)$ is obtained as the one that maximizes the Q-function $Q(s,a)$, i.e. $\pi(a|s) = \text{argmax}_a Q(s,a)$.

---

**Algorithm 2:** The toric code decoding game

---

Given: A policy network $N$
Initialize a new toric code state $s$ without errors
Add errors with probability $p_{\text{error}}$ per physical qubit
Measure the resulting *syndrome*
**while** *syndrome is not empty* **do**
 **foreach** *perspective $\mathcal{P}_i$ of $s$* **do**
  | Evaluate network $N(\mathcal{P}_i)$ to get move $a_i$
 **end**
 **if** *training* **and** *best action $a_i$ already taken* **then**
  | terminate and send reward 0
 **end**
 Execute best $a_i$, update $s$
**end**
Evaluate total error string (including correction)
Reward $= +1$ if no non-trivial error-string, else 0

---

Pauli operator to act on it with. Acting on qubits with no adjacent syndrome is not useful in this scenario however, and combined with the periodicity of the toric code this means we can use the idea of perspectives from Ref. [14] to restrict the output size of our policy network to just 12 actions: which of the four neighboring qubits of a centered plaquette/star to act on, and with which Pauli operator. Hence as input to our network we don't use a single version of the game state, but we create different views (perspectives) of the game, one for each syndrome defect, in which that syndrome is shifted to a central reference location. Finding the best overall move can then be performed by finding the move with the highest probability among all the perspectives. A limitation of our approach is that the probability of taking an action is not normalized over all perspectives. Contrary to Q-learning where the best action is unambiguously the one with the highest Q-value returned by the Q-network, here the output of the policy network indicates the best action for a single perspective *independently* of all the other perspectives generated from the same error sample. Algorithm 2 shows pseudo-code for the game steps.

For bit-flip only noise, the input dimension is $d^2$ since we only need to measure the values the plaquette operators. The output space can then be reduced to 4 actions corresponding to applying a Pauli-X operator on one of the four qubits neighboring the centered plaquette.

Our approach is not biased towards selecting the smallest error-correcting chain [14, 15], but is aimed at learning the most probable error strings (given a syndrome) like maximum-likelihood decoders [4, 10, 11], since our reward is only a function of whether there is a logical error in the final state or not (and is hence obtained only at the end of a game). During training, the game also ends in a loss if the agent decides to repeat an already chosen action. For the performance evaluation of a given neural network, we instead allow the same action to be taken twice and limit the game through a maximum number of decoding steps. If this maximum number of steps is reached, the game is lost.

## 4 Results

A common way to measure the performance of a decoder is to track the logical fidelity, i.e. the probability of introducing a logical error, against the physical error rate $p_{\text{error}}$. This quantity is

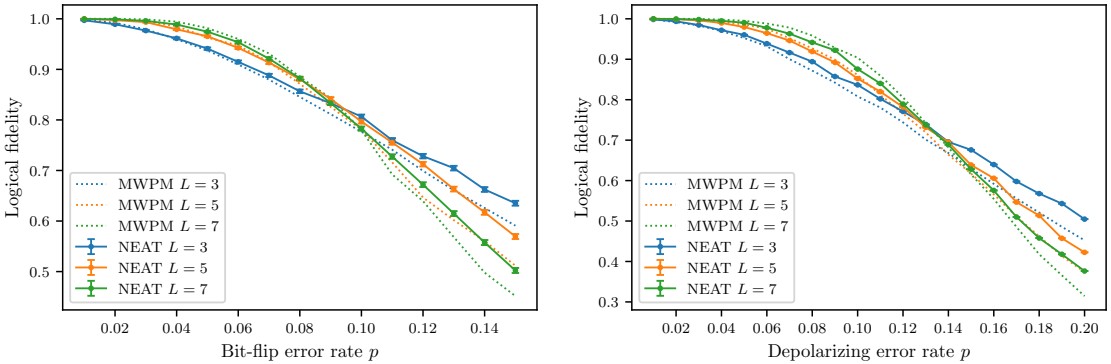

Figure 3: Logical error probability as a function of physical error rate $p_{\text{error}}$ for different code distances $d$, for bitflip noise (left) and depolarizing noise (right). The results of MWPM are shown in dashed lines. The curves show the best performing policy network found by NEAT. Evaluation of the logical fidelity is done on $10^4$ independent random games for each physical error rate.

computed as the ratio of successfully decoded cases (reward $+1$ returned from Algorithm 2) over the total number of games played, corresponding also to the fitness of our networks. Fig. 3 shows the performance, for both types of noise, of the best neural-network found by the NEAT algorithm after a few hundreds of generations. The optimization was stopped when the performance of the best network in the population saturated, for which typically about 600 generations sufficed.

An important advantage enabled through the genetic encoding of NEAT is that of being able to transplant genomes from smaller code distances to larger distances, as described in Appendix B. The policy networks for $d = 5$, for example, were initialized using the best genomes from the optimized $d = 3$ runs, speeding up the resulting optimization for $d = 5$.

The decoders found by NEAT have an error threshold. For both types of noise, the performance deteriorates as the physical error rate is increased and a crossing of the curves is visible around $p_c \approx 0.08 - 0.09$ for bitflip noise and $p_c \approx 0.13 - 0.14$ for depolarizing noise, which is a little worse than MWPM with $p_c \approx 0.11$ for bitflip noise and $p_c \approx 0.15$ for depolarizing noise [11, 12]. Nevertheless, the logical fidelity is slightly greater than MWPM for the largest error rates beyond $p_{\text{error}} = 0.1$.

We expect these differences to be due in large part to the absence of fine-tuning of the weights, because we observe that performance saturates during the evolution. It could be, however, that (much) larger networks are required for further small improvements to the threshold. The NEAT algorithm is not designed to find large networks, though extensions (such as hyperNEAT [28]) and other genetic algorithms can optimize large-scale neural networks [21].

In practice, for this work, we run NEAT separately for different code distances $d$. We point out that this makes the algorithmic error threshold somewhat ill-defined, in principle, since the decoders for different distances are not necessarily constrained to converge to the same decoding algorithm. The hyperparameters we chose for the mutation rates are reported in Table 2 in the Appendices.

We are able to reach the same performance as previously reported with RL methods [14–16] (we note Ref. [15] obtains higher error threshold and fidelity on depolarizing noise), though these results are obtained with considerably smaller neural-network decoders. Indeed, as can be seen in Table 1, our policy neural networks have three to four orders of magnitude fewer parameters than the deep Q-networks used in Q-learning, though it should be noted

Table 1: Number of parameters of the deep Q-networks and of the policy-neural-networks found by the NEAT algorithm.

| Decoders | Noise | $d = 3$ | $d = 5$ | $d = 7$ |
|---|---|---|---|---|
| [14] | Bitflip | | $\sim 500000$ | $\sim 1200000$ |
| [15] | Depolarizing | | $\sim 900000$ | $\sim 9000000$ |
| [16] | Bitflip | $\sim 640000$ | $\sim 1700000$ | $\sim 3200000$ |
| [13] | Bitflip / Depolarizing | | $\sim 2000000$ | |
| NEAT | Bitflip | **32** | **63** | **129** |
| NEAT | Depolarizing | **203** | **562** | **1188** |

that Ref. [13] deals with faulty measurements, which is a considerably harder decoding task. We also remark that we did not investigate the number of required parameters for a policy network that is trained to mimics these Q-networks. Our results were obtained without the use of spatial information that comes with using convolutional neural networks as in Refs. [13–16]. We remark that accessing larger code distances still becomes increasingly difficult due to slow convergence, and the genome transplantation procedure was crucial in particular for depolarizing noise.

Fig. 7 in the Appendix shows the NEAT optimized policy-network for $d = 3$.

## 5 Discussion

In summary, we showed that the NEAT algorithm can produce a policy network that results in a decoding performance similar to MWPM and other RL approaches based on Q-learning. The NEAT algorithm has the further advantages of being easily parallelizable, it automatically finds the smallest networks, and is gradient-free. We are hopeful that we can extend these preliminary results to larger system sizes, in particular through genome transplantation that allows starting the evolution with a good initial population. Crucially, by performing optimization directly in policy space and thanks to the properties of the NEAT algorithm, we were able to achieve the decoding task with very small neural networks, which represents a gain of the order of $10^4$ in terms of number of network parameters. Our work shows that very shallow feed-forward networks are expressive enough to decode the toric code on bitflip noise though more depth might be needed to get better performance on depolarizing noise.

It would be interesting to see how these performances translate to harder decoding scenarios such as fault-tolerant computations. Allowing NEAT to evolve neural networks other than feed-forward could be a possible direction for improvements. Extensions of NEAT include the evolution of convolutional [29] or deep [21] neural networks. Performance enhancements could also be expected from the use of policy gradient methods [30], which may be used to further improve the weights in a network topology that was found using NEAT. Preliminary work using the hyperNEAT algorithm of Ref. [28] did not prove conclusive, although in principle hyperNEAT would allow one to discover and exploit the symmetries of the problem in an automatic manner.

All-in-all, we believe that the NEAT algorithm, and evolutionary strategies in general, provide a competitive and conceptually simple alternative to training deep networks for reinforcement learning [21].

All of the code used to produce these results is publically available in the accompanying GitHub repository: `https://github.com/condensedAI/neat-qec`.

## Acknowledgments

We acknowledge fruitful discussions about this project with Mats Granath. HT was supported by grants from the Fondation CFM pour la Recherche and from the Erasmus+ program of the European Union. This work also benefited from the support of the French Programme Investissements d'Avenir under the program ANR-11-IDEX-0002-02, reference ANR-10-LABX-0037-NEXT (projet AiQus). HT would like to thank the Niels Bohr Institute for hospitality during his stay. This project has received funding from the European Union's Horizon 2020 research and innovation program under the Marie Sklodowska-Curie grant agreement No. 847523 'INTERACTIONS', and the Marie Sklodowksa-Curie grant agreement No. 895439 'Con-QuER'. The numerical calculations were performed on the local cluster of the CMT group at the NBI as well as in CALMIP (grants 2018-P0677, 2019-P0677). The NEAT simulations were done using the neat-python library [31].

## A  Extra NEAT info

This appendix is aimed at adding extra details to the NEAT algorithm description in the main text. Nevertheless, we have eluded some technical details that we leave to the original reference [20].

**Genetic encoding and crossovers.**  Each neural network of the population is encoded by a genome as shown in Fig. 4a. The key insight of [20] was to introduce an innovation number that keeps track of the history of a gene. Every new connection appearing in the population (see Fig. 4b) via a mutation is assigned a unique identification number (note that weight mutation does not generate a new innovation number). This crucially enables a simple and meaningful procedure for the crossover of two neural-networks as shown in Fig. 4c.

**Protection of the innovation by speciation.**  Another key element of NEAT is the design of a speciation mechanism that allows subgroups of similar neural networks (i.e. species) to evolve separately from the rest of the population. When the architecture of a neural network is changed via a mutation, it is likely that it will not perform well at first and a few generations are needed so that its weights can be adjusted. The issue is that the selection rules will eliminate these more complex individuals and effectively prevent better topologies to be found. It is possible to circumvent this issue by creating niches of individuals that share characteristics among themselves but not with the rest of the population, and applying selection independently on these subgroups. As a result, speciation is able to protect genetic innovation.

In [20], the species are defined via a compatibility distance $\delta$ which simply accounts for the number of excess $E$ or disjoint $D$ genes between two genomes, as well as the average weight differences in the matching genes $\overline{W}$:

$$\delta = c_1 \frac{E}{N_{\text{genes}}} + c_2 \frac{D}{N_{\text{genes}}} + c_3 \overline{W}, \tag{2}$$

where the $c_i$ are hyperparameters and $N_{\text{genes}}$ is the number of genes in the largest genome. At each generation, genomes are sequentially placed in species by checking whether the compatibility $\delta$ between the current genome and a genome randomly picked from a given species is below a threshold distance $\delta_c$. Additionally, NEAT employs a heuristic called explicit fitness sharing which favors homogeneity inside the species. The idea is to fight against the tendency that largely-populated species take over the rest of the species. This works by adjusting the

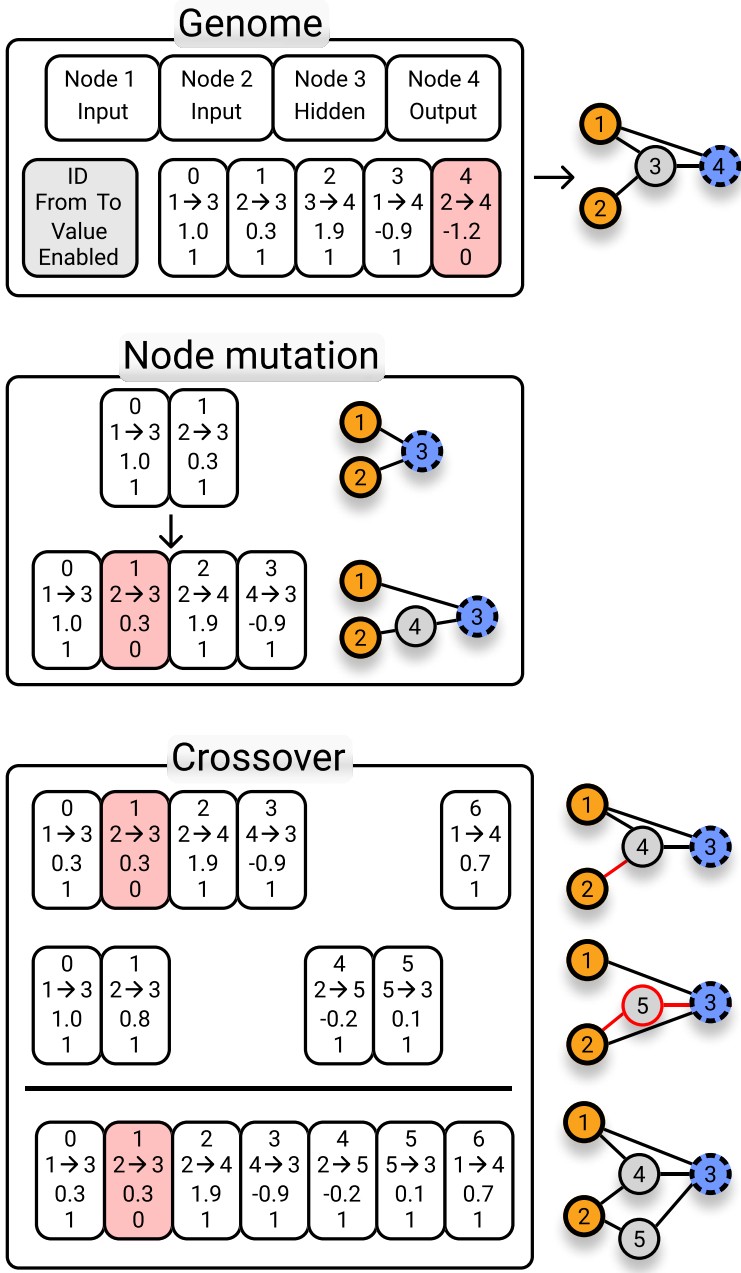

Figure 4: The genome of a neural network contains node and connection genes. A node gene stores an identification number and its type (input (sensor), hidden or output). A connection gene informs about which nodes it connects (the directionality allows to define recurrent connection that creates a loop in the neural network structure), the weight value it carries, a Boolean variable allowing for disabling the connection and, crucially, the *innovation number* (see main text). All this information uniquely define a phenotype neural network. Adding a node is done by splitting an existing connection in two, where the previous connection (here $2 \to 3$) is disabled and two new connection genes are created. Crossover is achieved by matching connection genes that share innovation numbers between the two parents (here genes 0 and 1). These matching genes are transmitted to the offspring with a weight and disabling option that is picked with equal probability from one of the two parents. The other disjoint genes are inherited randomly by the offspring.

size $N_j$ of species $j$ according to the ratio:

$$N_j' = N_j \frac{\overline{f_j}}{\overline{f}}, \tag{3}$$

where $N_j'$ is the size of species $j$ for the next generation, $\overline{f}$ is the fitness averaged over the entire population and $\overline{f_j}$ averaged over the individuals in species $j$.

**Minimizing dimensionality.** The last key insight of [20] is to initialize the population with neural networks having the simplest topology possible. For instance, neural networks of the first generation have no hidden nodes. In combination with speciation, this is argued to minimize the complexity of the final solution. Indeed, new architectural components are tested and optimized independently thanks to speciation: if the architectural innovation is proven to provide a significant performance boost, it is then included in the rest of the population. This way the complexity of the population only increases when necessary. Starting the evolution with the simplest neural networks hence ensures that the final solution has minimal complexity.

## B  Genome transplantation

Because of the perspectives, it is possible to transfer a decoder trained on a small code to a larger code. This can be done by performing *genome transplantation*, creating a network $N_2$ for code distance $d_2$ starting from a network $N_1$ for distance $d_1 < d_2$. This works by adding $2(d_2^2 - d_1^2)$ new input neurons to $N_1$ that correspond to new plaquette and star operators. All the weights connecting these neurons are set to 0, effectively ignoring the region beyond a distance of $\frac{d_1}{2}$ from the (reference) center. An example resulting transplanted neural network is showed in Fig. 5 with $d_2 = 5$ and $d_1 = 3$ with $N_1$ being the neural network shown in

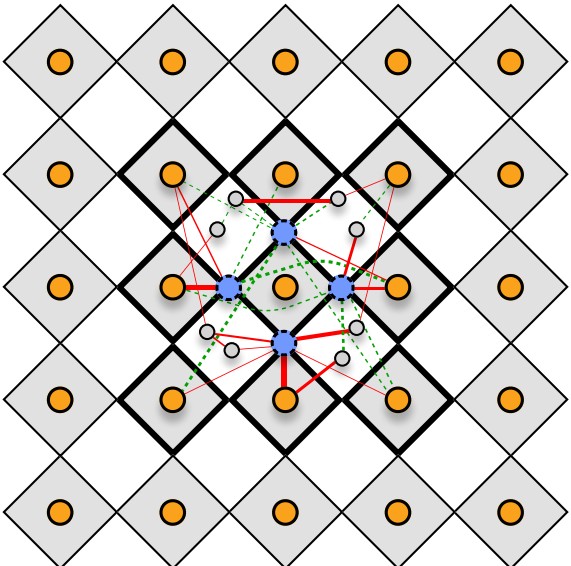

Figure 5: A neural network obtained from training at $d = 3$ (see Fig 7) can be used as a $d = 5$ decoder by inserting plaquette input nodes without connection weights linked to the rest of the neural network.

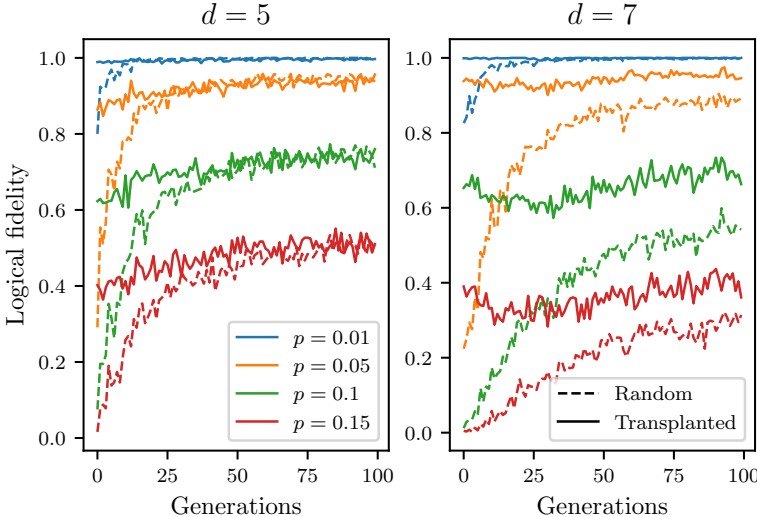

Figure 6: Logical fidelity against training time (measured in number of generations) for the best individual of each generation evaluated on 1000 random syndrome configurations at physical error rates $p = 0.01, 0.05, 0.1, 0.15$. This is a typical training run for the bitflip noise model. The dashed lines correspond to starting the training procedure with an initially random population, while solid lines correspond to starting with a population of *transplanted* neural-networks from the best $d_1 = 3$ decoder for $d_2 = 5$, and the best $d_1 = 5$ decoder for $d_2 = 7$.

Fig. 7. Fig. 5 also shows the performance of such transplanted genomes starting from a neural network trained at $d = 3$.

In the limit of small error rates, as can be seen in Fig. 6, the *transplanted* decoders perform well. This can be explained by the fact that in that limit there are only a few errors, each separated by a distance that grows on average with code distance, therefore the fact that the transplanted neural networks ignore long-distance information does not affect performance in this error regime. Fig. 6 shows that genome transplantation can accelerate the training quite significantly, in particular for the largest system sizes.

## C   Training hyperparameters

The population of neural networks in our runs varied from 100 individuals up to 300 for the largest code sizes. Each neural network initially has no hidden nodes but is fully connected from the input layer to the output layer, i.e. every input node is connected to every output node. The initial values of the connection weights and node biases are sampled from a Gaussian distribution with zero mean and unit standard deviation. The activation functions are all chosen to be sigmoidal.

During training, at each generation, the fitness of each neural network is evaluated on a set of 400 puzzles (500 for depolarizing noise) of varying difficulty, obtained from generating errors at $p_{\text{error}} \in \{0.01, 0.05, 0.1, 0.15\}$ ($p_{\text{error}} \in \{0.01, 0.05, 0.1, 0.15, 0.2\}$ for depolarizing noise) in equal proportion. In addition to that, to keep track of the best neural network over all generations, we evaluate the best-performing one from each generation on a separate dataset of about 5000 puzzles, which was generated independently at generation 0.

The mutation rates and other relevant hyperparameters are listed in Tab. 2.

As input data we have chosen to use the $2d^2$ values of the stabilizers $P$ and $S$. Alternatively,

Table 2: Mutation rates

| Hyperparameter | Value |
|---|---|
| Add/remove connection rate | 0.1 |
| Add/remove node rate | 0.1 |
| Weight mutation rate | 0.5 |
| Bias mutation rate | 0.1 |
| Enabling/disabling mutation rate | 0.01 |

one can also include the values of the physical qubits – except for the actual error chain – (projections along $z$ and $x$ axis), increasing the input size by a factor 3. Provided with the information about qubits, the agent effectively has memory of the past (it can see whether a Pauli X or Z operators has been applied already) and one may expect that this will improve performance. However in practice we were not able to find better decoding strategies with memory; rather, we observe slower training and convergence (in terms of CPU time) due to the larger networks. We believe that these limitations could originate from the NEAT algorithm, displaying slow convergence for the optimization of large networks in general.

Regarding the output, we also investigated the implementation of rotation invariance in the perspectives. This allows a reduction of the number of output neurons to 3 instead of 12, which corresponds to acting with the three possible Pauli matrices on a single reference qubit. The perspectives then contain the translated copies of the toric code but also the four rotated views for each of these, which effectively implements rotation invariance. Here again, we find that this trick did not improve performance. Instead, we observed that NEAT gets trapped more easily in local minima.

## D   Example NEAT network

Figure 7 shows an example network that was evolved using NEAT for $d = 3$ with bitflip noise, superimposed on top of a slightly different representation of the toric code.

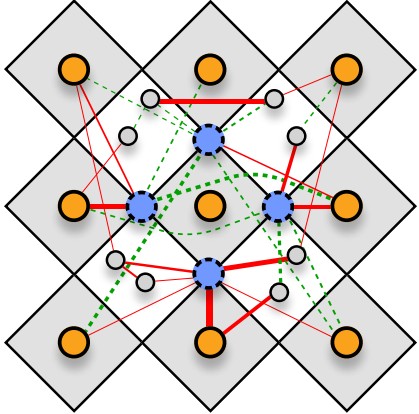

Figure 7: Architecture of a $d = 3$ NEAT decoder for bitflip noise, rotated with respect to Fig. 1 for convenience. The neuron inputs are placed where they are located on the lattice, as are the four outputs. The width of the edges are proportional to the corresponding absolute value of the weights. Positive (negative) weighting is shown with dashed green (solid red) lines.

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
