# Peer review of "A NEAT Quantum Error Decoder"

_SciPost Physics, doi:SciPost Phys. 11, 005 (2021)_

## Round 1 · Referee Report · Anonymous (Referee 1) · 2021-4-20

Report

The paper by Théveniaut and van Nieuwenburg presents a neural network based decoder for the toric code built on the NEAT algorithm. Similarly to recent decoders based on deep reinforcement learning (DRL), the neural network takes as input an error syndrome and outputs a best qubit action for error correction. However, here, instead of using standard backpropagation and gradient descent to train the network, an ensemble of networks is evolved in an evolutionary fashion through genetic mutations that change weights or merge networks. I find the results novel and interesting, and the presentation is very accessible both of the decoder and the NEAT algorithm. In terms of results the performance is not very impressive, the NEAT decoder does not outperform standard minimum matching, but the paper is more exploratory in nature and presented as such. As one of many different machine learning approaches to decoding topological codes, this is certainly one of the most original, with potential to be developed further.

I have a few comments, of which I think the first one needs to be addressed properly.

1) There is too little details about the training (evolution) of the networks. What is the convergence criterion?

If I understand correctly the reward is formulated in a way that should in principle give an optimal (maximum likelihood) decoder (in contrast to what’s stated in the paper below algoritm 2): The scheme takes as input a randomly generated error chain, based on the corresponding syndrome the decoder suggests a correction chain, and the complete chain is evaluated with respect to non-contractible loops corresponding to logical operators. A decoder that learns to maximize the return based on this reward should be optimal. (Although it may be difficult to converge since the reward signal will be obscured by the statistical fluctuations that follow from using randomly generated error chains which may or may not fall in the most likely equivalence class.) In the paper it is stated that the results are based on “best neural-network found by the NEAT algorithm after a few hundreds of generations”. What does this imply in terms of well the network actually manages to maximize the return? Given the suboptimal performance, especially for depolarizing noise, it seems likely that the evaluated networks are quite far from maximizing the return. This may also have implications for the small size of the networks as shown in Table 1. Wouldn’t they continue to grow if a more stringent criterion for the training convergence was used? How large the training set of syndromes is would probably also have implications on this growth of the network size.

2) I find that there is some confusion in the presentation about what code is used. In the actual calculations (as shown in Fig. 5 and 7) the standard toric code is used. In Fig 2 is shown the rotated code. If this is put on the torus as in Fig 2, the space of logical operators are not as neatly defined as for the standard toric code where logical X and Z are put on alternating bonds. I don’t think there is anything wrong in principle here, but it may have been better to stick to the standard toric code representation.

Also, for information, Figure 1 and 2 does not seem to display properly on Safari and Preview.

3) When referring to maximum likelihood decoders I suggest that the authors also cite the Markov chain Monte Carlo based decoder: J. R. Wootton and D. Loss, Phys. Rev. Lett. 109, 160503 (2012). 

Typo: There is an extra “of”, first sentence of section IV.

  • validity: -
  • significance: -
  • originality: -
  • clarity: -
  • formatting: -
  • grammar: -

Author:  Everard van Nieuwenburg  on 2021-06-03  [id 1482]

(in reply to Report 1 on 2021-04-20)

Dear Referee #1,

We are very appreciative of your clear understanding of what we hoped to achieve with this manuscript. As you state, our aim was not to construct a decoder that outperforms others, but rather to explore optimization of a DRL agent using a genetic approach that has proved very successful in other RL problems (notably, on game playing environments). We do expect however, that further inclusion of gradient-based updates will contribute to a better performance of the decoder, perhaps even pushing it considerably closer to the minimum weight perfect matching threshold. We believe it was worth leaving them out, to demonstrate that mostly random weight mutations still manage to get a good performance even in the absence of fine-tuned gradient steps.

The missing convergence criterion is a serious omission on our side, and has now been remedied explicitly in the text (in section IV, and in appendix C). In addition, we are now linking to a github repository that contains the full source code with our implementation.

"If I understand correctly the reward is formulated in a way ..."

The referee is absolutely correct in this understanding, and we have corrected this mistake in the manuscript. The reward scheme is indeed such that the resulting decoder should in principle be a maximum likelihood decoder.

"In the paper it is stated that the results are based ..."

With “the best neural network found by NEAT” we meant to indicate that we are picking the (single) network from the population that has the highest fitness. That is, the network that solves the most out of a set of N random syndromes correctly. By growing this set of test-games, we can increase the resolution we have in terms of the performance, though this makes sense only if many of the networks are able to maximize the score. We concluded that the remaining discrepancy is mainly due to the impossibility of identifying the correct error string, rather than an issue with the network sizes. However, the referee is correct in that NEAT is essentially biased to finding small networks. If a large network is required, it may take a long time for NEAT to find it. This consideration is what is behind the hyperNEAT algorithm, which moves towards the evolution of very large networks (see e.g. refs 18 and 21 in the paper). In hyperNEAT, large neural networks are encoded through small neural networks representing the genome (i.e. they take as input 2 neuron IDs, and predict the weight connecting them). We now comment on these considerations in the results section of the manuscript.

"2) I find that there is some confusion in the presentation about what code is used. In ..."

This choice was made initially purely for aesthetic reasons, but we would rather avoid confusion in the first place. We have amended this and are now using the standard toric code also in Fig 2.

"Also, for information, Figure 1 and 2 does not seem to display properly on Safari and Preview."

Thank you for letting us know. We were able to reproduce this issue, and have since switched to a different method of generating the figure PDFs that seems to resolve it for us.

"3) When referring to maximum likelihood decoders I suggest that the authors also cite the Markov chain Monte Carlo based decoder: J. R. Wootton and D. Loss, Phys. Rev. Lett. 109, 160503 (2012). 
"

This reference, along with a two other missing ones, has now been included. Thank you for pointing this out.

"Typo: There is an extra “of”, first sentence of section IV."

Fixed!

---

## Round 1 · Referee Report · Anonymous (Referee 3) · 2021-4-22

Report

After a discussion with the editor-in-charge, due to my initial delay accepting this review request, I am only submitting short general opinion, not a full length report:

I find this manuscript represents a fascinating work. When using neural networks for physics motivated tasks we are generally lacking on good optimisation methods for their architectures. The introduced method, based on genetic optimisation, works very well and appears to scale very well. The introduced method is original and creatively addresses one aspect of scalability of error correction decoders. I am very impressed with the result!

---

## Round 1 · Referee Report · Anonymous (Referee 2) · 2021-4-22

Strengths

  1. Using the NEAT algorithm, the authors can find much smaller neural networks compared to previous works. This can lead to faster decoding time.

Weaknesses

  1. Not clear how to scale to larger distance codes and multiple rounds of measurements

Report

This work demonstrates that it is possible to automatically find smaller networks to decode surface codes compared to manually designed ones. I find it satisfies the acceptance criteria. (I'm not a good judge of writing and formatting, so I won't comment on these aspects)

I have a few questions for the authors. Please at least add some comments about question 1 and 2 in the paper. 1. In recent years, there are a lot of works on AutoML (https://en.wikipedia.org/wiki/Automated_machine_learning, https://en.wikipedia.org/wiki/Neural_architecture_search). It also cares about finding neural network structures automatically. Can you comment how Neat compared to some newer methods in AutoML?

  1. Why is the gradient-free aspect of NEAT considered as an advantage? In the recent years, a main reason to use neural networks is that they work so well together with gradient optimization.

  2. A related question is, since NEAT doesn't need gradient optimization, have you considered to use some other machine learning models such as decision trees instead of neural networks?

  3. I feel a good way to scale-up might be concatenate your neural decoders with another easily scalable decoder. For example, see https://arxiv.org/abs/2101.07285. Will this be possible? Do you think the end result can be better than this work?

  • validity: high
  • significance: good
  • originality: high
  • clarity: good
  • formatting: -
  • grammar: -

Author:  Everard van Nieuwenburg  on 2021-06-03  [id 1483]

(in reply to Report 2 on 2021-04-22)

Referee #2 asks on-point questions that we will elaborate upon here (and in the updated manuscript). We thank the referee for raising these points, which we have used to improve our manuscript, and for putting our work into perspective.

The aim of our manuscript was not to provide an algorithm that scales to much larger systems, though indeed smaller networks (fewer weights) might be beneficial. A better approach to scaling, as the referee suggests, would then be to combine small decoder networks with more easily scalable decoders as in Meinerz et al. (this reference has now been included!). Possibly, other approaches such as the the hyperNEAT algorithm (Refs 21 and 30 in the manuscript) are of interest, since they can optimize large networks (which NEAT is not very good at).

The gradient-free optimization has a few properties that can be advantageous that we wanted to explore. Namely, apart from not requiring gradient computations at all, it requires much less hyperparameter tuning in practise than standard gradient descent optimizations of neural networks (See ref 21). Second, gradient-free optimization methods are less prone to get stuck in local minima and generally work better in rugged loss landscapes: They even typically work well for non-smooth or discontinuous loss landscapes, which may be an advantage for learning a (discontinuous) policy. This is also relted to the observation that genetic algorithms can work well for large-scale networks, and can succeed in cases where other agents, in particular DQN agents, fail (also in ref 21).

We did not consider the use of other non-gradient based methods (most notably CMA-ES), because we wished to focus on neural network based decoders. The genetic method for searching for neural network topologies is actually part of the AutoML concept, mentioned under the wikipedia page linked to by the referee. The other dominant approach in the spirit of AutoML is using reinforcement learning to adjust the topology, which we expect would have been more costly in computational time (not as easily parallelizable).

---

## Round 2 · Referee Report · Anonymous (Referee 1) · 2021-6-17

Report

As stated already in my first report I think this is a very interesting and novel approach to decoding topological error correcting codes. The paper is very accessible, clearly structured and presented. With the modifications made in response to the referee reports I recommend that it be accepted to SciPost Physics. I agree with the assessment of the 3rd referee that this is a top 10% paper.

---

## Round 2 · Author Response

Dear Editor,

This new version of the manuscript has an improved presentation (notably the consistency of Fig. 2 with the ones in the appendices).
Both (pre-editorial recommendation) referees raised excellent in-depth points (for which we are very grateful), and we believe that we have addressed these points in the current version. The third post-editorial recommendation report is very motivating too. Replies to the referees will follow separately.

We have now also explicitly included a link to our github repository that contains all of the code to reproduce our results.

Thank you for your effort and patience in getting our work reviewed.

---

## Round 2 · List of Changes

Changes and additions to text and figures: * Added sentences about the convergence criterion for our runs in section IV and in appendix C. * Add link to github repository hosting all code for reproducibility * Updated figure 2 to show the non-rotated code * Updated text underneath Algorithm 2 to reflect that we do, in fact, expect to get a maximum likelihood decoder (Thank you @Referee #1) * Added paragraph in Results section discussing the discrepancies between the NEAT decoder and MWPM

New citations: * J. R. Wootton and D. Loss: https://arxiv.org/abs/1202.4316 * Meinerz et al.: https://arxiv.org/abs/2101.07285 * Chamberland and Roonagh: https://arxiv.org/abs/1802.06441

Fixed typos: * Extra “of” in the first sentence of section IV (Thank you @Referee #1)

---

## Editorial Decision

published